# "And You Became Mine" (Ezek 16:8): Good and Evil in a Narcissistic God

## Gili Kugler

The Department of Jewish History and Biblical Studies, The University of Haifa, Haifa 3498838, Israel; gkugler@univ.haifa.ac.il

**Abstract:** Scholars have defended the cruel behavior of the biblical god as being justified, due to the supposition of God being perfectly omnipotent and infallible. However, one cannot be obtuse to the depictions of Yhwh himself about his feelings and actions, as expressed through biblical narratives and brought forth by his emissaries, the prophets. When observing the prophecies of Ezekiel, for example, through a modern psychological lens, God's relationship with his subjects, and especially with his offspring—the nation of Israel—reveals clear patterns of malignant narcissistic behavior. This study proposes that evil is an immanent part of God's nature in the Hebrew Bible. The texts make no effort to disguise God's narcissistic nature in his behavior towards his chosen one, a behavior that resonates with patterns one would define as evil. Moreover, the texts reflect the willingness of followers to acknowledge their situation as being trapped in an abusive relationship with a vicious patron.

**Keywords:** evilness in God; biblical theology; familial imagery; divine incest; narcissistic personality disorder; narcissistic parenting; Ezekiel 16

## 1. An Epistemological Inquiry

Does God intend to be good? Or is he actually evil? Is it even possible to determine and define good or evil nature? Like the elusive notion of *God*, so are the concepts of *Good* and *Evil* fluid and relative; alternating and influenced by the observers. One may see a powerful entity inflicting harm upon a helpless party and consider this action as evil. However, such a judgment would have a cognitive bias, as the observer is always ignorant of the full circumstances in detail and of the background of the involved parties, along with their drives and motives. Often, what is perceived by one as a sharp asymmetry of power is judged differently by another. Additionally, in the case of the participants, even a supposedly stronger side would almost never see themselves as being guilty or evil. Russian soldiers in Ukraine do not perceive themselves as the "bad guys", nor do ISIS fighters, or those who stormed Capitol Hill. Nor does any dictator that oppresses his subjects.

When defining the relative nature between *good* and *evil*, one must also consider the influential aspects of cultural and temporal factors. What today would be considered abusive or immoral, like a parent beating their child, would be, in other times, a manifestation of positive parenting, as reflected in the biblical book of Proverbs. Similarly, actions that are nowadays considered crimes against humanity, such as perpetrating destruction on cities and nations, have been depicted in the biblical text (e.g., Genesis, Exodus, Numbers) as a positive requisition by a supreme, powerful, and sometimes divine authority, who then advises his people to act accordingly (e.g., Deuteronomy, Jushua, 1Kings) in what today would be considered severe heinous behavior.

Yet, although *good* and *evil* are entirely relative, varying through cultures, generations, and perspectives, there have always been attempts to define them, or trace their essence. Such attempts have been especially ambitious considering theistic presuppositions of the

existence of a supposedly moral role-model for humanity in the form of an omnipotent deity. Solving the contradiction between the existence of evil beside the omnipotent, omnibenevolent, and omniscient god has occupied minds for centuries, and has produced various arguments in the theological and philosophical domains.[1] James Sterba's recent book revisits the question from a logical-philosophical perspective, deliberating the presumption of an "all powerful, all knowing, and perfectly good god", against the universal human experience of evil. Sterba finds it logically difficult to reconcile a traditional understanding of God's perfection with the presence of evil. He contends that an attentiveness to the horrendous moral and natural evils in the world cannot help but undermine belief in the traditional deity, in whom the virtues of omnipotence, omniscience, and moral goodness are thought to coincide.[2] Sterba's discussion responds, inter alia, to the endeavors of scholars from the monotheistic faith, to reconcile the conviction of an omnipotent deity—an ideal role model for men and women, created in his image, with the existence of evil. Theists dealing with theodicy have employed various defenses and doctrines, such as concepts of moral retribution and the afterlife, to usually decline the likelihood that a supreme deity allows wickedness, or even produces it.[3] A supposed existence of evil in the world has often been explained in accordance with monotheistic axioms, which entail pedagogical presuppositions as to the role of God for humanity and morality.[4]

Nonetheless, the attempts to associate omnipotence with goodness do not represent all monotheistic paradigms. The Hebrew Bible [HB]—a very first foundation of monotheistic beliefs and denominations, does not seem to be concerned with a possible association between the deity and depravity. Second Isaiah, for example, being triggered by the Zoroastrian religion, does not hesitate to declare that God himself is the source of evilness, as much as he is the derivation of wealth, life, and death (Isa 45:7). Could this idea imply that God not only creates or allows evil, but also contains wickedness? This question is not about the very existence of God and his actual traits, but about the way he is portrayed and perceived by humans.

While the HB contains complexities and discrepancies in the depictions of the deity, it also reflects different sets of expectations and values than those envisioned by later readers and followers. The question of this study moves from the ontological realm to the epistemological one. Rather than asking whether God is good or wicked per se, we ask whether he was *portrayed* as such. In the following sections, we will delve into one of the first foundations of monotheistic thought, the Hebrew Bible, to argue that the depictions of God do not presuppose goodness as a necessary characteristic of the deity.[5] This study will show that whether the biblical god was considered omnipotent or not (this is a whole other issue in the examination of God's nature)—evilness has been an immanent trait in representations of the deity.

## 2. "God" as a Product of Scribes and Society

Anthropomorphic imageries in the HB deliver a human-like portrait of the biblical deity. Hence, an examination of the deity's nature ought to be conducted by the same means used upon humans. Defining "goodness" and "evil" in human nature mandates an evaluation of relationships between parties and their attitude towards each other, thus revealing intentions of good will or inclination to cause harm.[6] This study argues that a reliable judgment of the image of the biblical god may only be obtained through an examination of his relationships with humans, and especially through his relationship with his chosen kin, Israel. This chosen kin, according to the diverse theological myths of the HB, is the one whom God singles out, saves, adopts, and takes as a wife. Thus, Israel is the ultimate candidate upon whom God's relationships can be comprehended. In some cases, it is God himself who reveals a first-person view of the relationship, by declaring his aim and purposes in maintaining the relationship.

This study departs from the philosophical discussion and moves to the literary realm, offering an analysis of the concept of an omnipotent god, as provided in one of the imageries of the HB. While cultural gaps and relative perspectives make it difficult to define

"evil", the texts in hand reflect the perception of the scribes that good or evil coexist in the concept of *God*. This is revealed through an exposure to the deity's inner thoughts and intentions regarding people and nations. The texts we shall see indicate the authors' readiness to define the deity as performing merely in his own interest, and not for the "good" of others. The recipients of these texts, and the society in which the texts were produced, are familiar with such characteristics, and are willing, at a certain point, to embrace them and canonize them. Unlike voices heard today about God, it seems that the authors of the text, as well as their community, did not refrain from imagining their deity as containing *evilness*; even if not stating it directly.

### 3. For the Sake of God's Name

When reading the illustration of God's association with his chosen kin, Israel, in the book of Ezekiel, the reader in confronted with the protruding attribute enveloping God's behavior—acting for the sake of God's name. This eccentric choice is found in the different forms of the text, whether in historiographical reviews, prosaic statements, or poetic prophecies. It brings forth the complexity of the relationship between Israel and the deity, and reflects presuppositions and beliefs in regard to God's intentions and morality. Can God's choice to act for his name be described as pride and arrogance, and tainted by an evil root in his nature?

Prophecies in the book of Ezekiel relate a significant theological role to the name of God. First-person statements in the book convey the idea that God performs for the sake of his name, למען שמי ("for my name", Ezek 20:9, 14, 22, 44); a name through which he should be known, both by the nation Israel (12:16, 22:16) and by others (38:16, 39:6, 13, cf., v. 21). Further verses convey that the very destiny of his people, Israel, may affect the holiness of Yhwh in the world, e.g., ונקדשתי בם לעיני הגוים ("and manifest my holiness in them in the sight of the nations", 28:25, cf., 36:22–23; ואת שם קדשי אודיע בתוך עמי ישראל ("My holy name I will make known among my people Israel", 39:7, cf., 39:25).

While references to God's name are also mentioned by other prophets (e.g., Amos 2:7; Mal 1:12, cf. 2:11), the idea is predominantly found in the book of Ezekiel. Nonetheless, the BDB lexicon interprets Ezekiel's engagement with this notion of acting for the name of God as acting *according* to his attributes, as reflected in similar declarations in Jeremiah (Brown et al. 1906, p. 2496). In Jeremiah, the references to God's name relay the message that God acts or should act according to his divine traits ("למען שמך", Jer 14:7, 21), and namely as a people's savior and the keeper of the covenant (vv. 8, 9, 21), in spite of the apostasies of the sinner Judeans (vv. 7, 20). But in the theology of Ezekiel, the call of acting *for the sake of God's name* is not stated by the people. It is God's own announced ambition to be known and appreciated, and to be protected from profanation by any means (cf., 13:19, 20:39, 22:26). Unlike the suggestion of the BDB, the usage of God's name in Ezekiel aims to ensure that the reputation of God will be kept, shown, and glorified.

The principle of manifesting God's name, according to Ezekiel, is what saves the people from annihilation. Ezekiel 20 narrates that even though the people deserve to be obliterated, Yhwh refrained from abolishing them (20:14, 22), due to his concern that the situation may be misunderstood by other nations. Indeed, when God executes judgment against Judah/Jerusalem, he does it for all nations to see (Ezek 5:8, 39:21), like a father smacking his child in public.[7] However, the awareness of the other nations to the unfortunate situation of God's people has also led to an erroneous interpretation of God's capabilities. This had a negative impact on the name of God, similar to the "profanation" caused to the Sabbaths (20:13, 16, 21, 24, 22:8, 23:38), and the temple (23:39, 25:3, 28:18, 44:6, cf., 24:21). God had been at risk of becoming profaned due to the misinterpretation of his actions (e.g., 36:20–23).[8]

The worry about the integrity of God's name has therefore been the incentive to implement God's name within Israel—an application of a physical or substantial element of God's existence: "My holy name I will make known among my people Israel; and I will not let my holy name be profaned any more; and the nations shall know that I am the LORD,

the Holy One in Israel" (39:7). For the same reason, a restoration of the exiles, God's people, would take place in the land: " . . . Now I will restore the fortunes of Jacob, and have mercy on the whole house of Israel; and I will be jealous for my holy name" (39:25). The restoration is not a result of the people's virtues, and will not be done for their own sake. It will occur to prevent harm to God's name among the gentiles, and to implement validation of his holiness in their understanding (cf., 20:41, 28:25, 29:6, 36:21-24, 36, 39:27, 28). A similar message is stated regarding the restoration of the Judean kingship, which will not be the people's reward, but rather an opportunity to acknowledge God's actions throughout history (17:24). Similar is the case of retaliating against other nations, when God's biggest desire is to " . . . display my greatness and my holiness and make myself known in the eyes of many nations. Then they shall know that I am the Lord" (38:23. Cf. 25:14, 17, 30:8, 39:6, 21, 22, 25). What does the tenet of God's acting for the sake of his name indicate as to his attitude towards his kin, and what does it reveal in regard to the objectives and intentions of this relationship? We will discuss these questions further below.

## 4. The Deity of Ezekiel Diagnosed with Narcissistic Personality Disorder

As we can see, the concern about *God's name* in history and eschatology lies at the center of Ezekiel's theology. This concern constitutes the incentive and drive for God's relationships with others, including his close kin, and reflects characteristics such as selfishness, egocentricity, megalomania, and exploitation. These characteristics resonate with the traits listed as God's character in Richard Dawkins' 2006, *The God Delusion*. Dawkins denounces the "God of the Old Testament", for being "arguably the most unpleasant character in all fiction: jealous and proud of it; a petty, unjust, unforgiving control-freak; a vindictive, bloodthirsty ethnic cleanser; a misogynistic, homophobic, racist, infanticidal, genocidal, filicidal, pestilential, megalomaniacal, sadomasochistic, capriciously malevolent bully" (Dawkins 2006, p. 51). Dan Barker, a former evangelical pastor, based an entire monograph on Dawkins's quotation, to demonstrate how *God* is *The Most Unpleasant Character in all Fiction*.[9] But one phrase is missing from both Dawkins's character description and Barker's survey, that seems to me the most applicable to the question of evilness—that is God's *narcissistic personality disorder*.

Whether deliberately or not, biblical depictions and narratives reflect tendencies of social connections and the human psyche. Over the past half century, studies demonstrated the contribution of theories in psychology for better understanding the characters and interactions involved in biblical ethos and narratives.[10] In 2009, Donald Capps used the 1994 *Diagnostic and Statistical Manual of Mental Disorders* (DSM-4) to indicate manifestations of *narcissistic personality disorder* in the biblical deity.[11] Capps's usage of the 1994 DSM should be carried on to the updated DSM version (5, 2013), which summarizes *narcissistic personality disorder* (NPD) as "a pervasive pattern of grandiosity (in fantasy or behavior), need for admiration, and lack of empathy . . . present in a variety of contexts" (American Psychiatric Association 2013, p. 670). While a diagnosis of NPD requires the manifestation of at least five out of nine listed criteria, Capps argues for a diagnosis of a "complete" and acute narcissist in Yhwh, as he expects "no less of a perfect God". He thus traces all nine criteria in Yhwh's character; including grandiosity and self-importance, fantasies of unlimited success and power, jealousy, and lack of empathy.[12]

While the notion of *evil* is elusive and difficult to define, easier is to trace are behavioral and social tendencies, and to relate them to what one may consider wickedness or evilness. As stated by Jaycee Hallford, the psychological definition of narcissism lies as a "supplementary trait beyond evil; alongside aggression, egotism, revengefulness, hatred, selfishness, and sadism".[13] In the case of the biblical god, it could be attested that his behavior often reveals him as a *malignant narcissist*, as he is willing to abuse others, even those dear to him, in order to retain constant acclaim and admiration.[14] This diagnosis is attested in biblical stories that narrate the tendency of God to harm not only his enemies (e.g., Deut 7:18–24; Ezek 25:7–13), but primarily his beloved and chosen entities, whether sinners (e.g., Exod 32:10; Num 14:12; 25:11; 2kgs 17:18) or pious (Job 2:3). We will examine this through

the allegory provided in Ezekiel 16, where God's performance in his relationship with his chosen child clearly manifests his malignant narcissistic nature.

## 5. Assessing *Good* and *Evil* through Familial Connections—The Case of the Ezekiel Allegory

Our interpretation will demonstrate the dark picture portrayed in the tumultuous allegorical narrative in Ezekiel 16. A treacherous and violent relationship is unfurled between a narcissistic authoritarian entity, Yhwh, and the helpless dependent female child, Jerusalem. Yhwh, a patriarchal character, plays dual roles for the female entity: first a caregiver, then a husband.

The use of dual roles in depicting God's relationship with Israel is well-known in biblical prophecy. As stated in the words of Second Isaiah: "For your Maker is your husband, the LORD of hosts is his name . . . " (Isa 54:4. Cf. 62:5–11). A prophecy in Jeremiah, for example, shifts between father-son and husband-wife imagery (Jer 3:2-4), with allegations directed to Israel as a promiscuous wife, and longings for a trustworthy and reliable relationship through the image of a daughter: " . . . You have played the whore with many lovers; and would you return to me"; "Have you not just now called to me, 'My Father, you are the friend of my youth'" (vv. 1, 4).[15]

The allegory in Ezekiel 16 constitutes a similar duality in Israel's interaction with Yhwh. Nonetheless, rather than shifting between the father-daughter and the husband-wife models, indicating the elasticity of the relationship, the spousal phase is portrayed in the allegory as the final goal, the target of their relationship; whereas a child may leave the parent, a spouse is obliged to remain at his side (cf., Isa 46:4). In her new position, the girl is punished by Yhwh for allegedly violating expected restrictions that have come with the change in relationship. How is God's personality in this portrayal perceived and interpreted? Are the deeds towards his protégé considered good or evil? How were they regarded by those who produced the texts?

In a leading comprehensive commentary to the book of Ezekiel from 1997, David Block argued that what may seem to a modern reader as a "revoltingly bloodthirsty" God, who is "devoid of the most elementary compassion or decency", is only an erroneous interpretation of the biblical text.[16] Using the above-mentioned factor of relativism, Block underlines a gap between contemporary moral expectations and the norms/morality at the time of writing. He argues that since the covenant of Yhwh with Israel is depicted by using familial and marriage metaphors, God's conduct should be considered "logical and natural", as it manifests an acceptable response to a wife's infidelity.[17] Consequently, he asserts, one should refrain from judging the biblical deity as cruel or unjust, and God's wrath in the relationship "should not blind the reader to the prophet's fundamentally positive disposition toward the covenant" (Block, *The Book of Ezekiel* 1, 49). Therefore, according to Block, God's revenge and retaliation against the nation, mentioned in the allegory, are considered proportional to the depth of God's covenant love (Ibid., p. 14). To summarize, Yhwh should be judged as "a gracious and compassionate God, who not only rescued Jerusalem . . . the abandoned infant, from certain death", but also "marries her, and with unrestrained expressions of love he elevates her to the status of queen" (Ibid., p. 49).

Putting our own judgment aside, do we have tools to assess the validity of Block's suggestion of Ezekiel's god being portrayed as "gracious and compassionate"? It seems that while Block avoids judging the actions of the god of Ezekiel according to contemporary values, other presuppositions he holds come into play. Block assumes that Yhwh represents a moral and justifiable system accepted at the time of production, which had welcomed demands for obedience and compliance, as well as punishments and vengeance against one's own protégé. Indeed, the prophecies of Ezekiel do not contain an explicit criticism of God's behavior, nor define it as evil, or a source of evilness; but the prophecies do not reverberate God as conducting goodness, either. As we shall see, Yhwh of the allegory is driven by his own needs, and not by the needs of his adopted child/spouse. This correlates with the broader view of God in Ezekiel, acting for the sake of himself, with no considera-

tion for humans, including his very close kin, Israel. Traits such as these are rudimentary in NPD.

### 6. "In your Blood Live" (Ezek 16:6): Supposed Grace and Compassion in the Allegory

The allegory of Ezekiel 16 narrates that Yhwh notices Jerusalem at a crucial moment in her life, just after being born and then forsaken by her parents—a Canaanite father and a Hittite mother (Ezek 16:3).[18] In the first days of her life, Jerusalem was moribund, with her "navel cord . . . not cut . . . [and being not] washed with water to cleanse . . . nor rubbed with salt, nor wrapped in cloths" (v. 4).[19] The lack of parental care in that crucial time of her life prompted Yhwh to approach the girl with the following instruction: ואמר לך בדמיך חיי בדמיך חיי. רבבה כצמח השדה נתתיך ("and I said to you: 'in your blood live, in your blood live'. I have caused you to multiply as the bud of the field", 16:6b–7a₁; my translation). However, these words are rather vague in meaning, and may leave the reader perplexed about the actual action taken by Yhwh for the sake of the abandoned girl. Benjamin Kedar-Kopfstein, for example, considered the instruction as Yhwh's command to the girl to live "in [her] blood" and be rescued from death, but with no promise of relief from her excluded and liminal situation (Kedar-Kopfstein 1978, p. 246). Mary Shields asserted that the sentence does not refer to God's care, love, or compassion, "until the girl/woman exhibits the 'ornaments of ornaments'" (v. 7) (Shields 1998, p. 8). Christl Maier suggested that the text indicates that Yhwh simply orders the infant to sprout like a plant, with no actual action of rescue or care. Thus, the girl remains in a "liminal state between the open field and the human realm".[20] More bluntly, Aaron Koller recently argued that the statement does not indicate a bestowal of care and tender for the girl, and that the allegory does not at all evoke an image of adoption by Yhwh (Koller 2017, p. 410).

A completely opposite view was argued by Block, as part of his thesis regarding Ezekial's compassionate God. Block asserts that the statement בדמיך חיי communicates Yhwh's intention to fulfill a missing parental role for the girl. This idea is supported by the earlier supposition of Meir Malul, that the words בדמיך חיי echo Akkadian formal declarations of adoption, which clarify one's intention to rescue a child from a state of emergency and position them under new ownership.[21] Indeed, the following phrase in the statement of Yhwh emphasizes his role in the envision of the girl's thriving ("I made"): השדה נתתיך רבבה כצמח ("I made you multiply as the plant of the field", 7a₁).[22] As such, the statement בדמיך חיי responds to the deprivation the girl had experienced in the beginning of her life, when "no eye pitied you, to do any of these things for you out of compassion for you; but you were thrown out in the open field . . . " (v. 5).

At this point, we can agree with Block's interpretation that the allegory depicts a scenario in which Yhwh steps into a missing parental role for the girl. This, however, cannot make one concur with Block's other supposition regarding to God's personality as gracious and compassionate. Instead, as we read the rest of the allegory through a *parental relationship* framework, God's vicious attitude towards the child is enhanced.[23] As we shall see, instead of functioning as a supportive and protective parent, Yhwh inflicts upon the child violent chastisements as a medium to first and foremost satisfy his own needs and ambitions.

### 7. "And You Became Mine" (Ezek 16:8): Exploitation instead of Grace and Compassion

We presented Block's assertion, that Yhwh's stepping into a parental role, as given in Ezekiel 16, supposedly manifests acts of grace and compassion. This assertion is problematic, as a further reading of the allegory reveals that God's gracious propensity does not sustain for long, as it serves a mainly self-interested goal.

After Yhwh saves the girl Jerusalem, he recognizes her reaching puberty and womanhood—the "age for love" (והנה עתך עת דדים, v. 8). Consequently, he updates his intentions towards the girl and makes her available to him as a developed woman: "You grew up and became tall and arrived at full womanhood; your breasts were formed, and your hair had grown; yet you were naked and bare. I passed by you (again) and looked

on you; you were at the age for love. I spread the edge of my cloak over you, and covered your nakedness" (vv. 7a₂–8a). Upon recognizing the girl's feminine nature, Yhwh covers her nakedness. This act, rather than expressing care, conveys the enforcement of the new authority upon the girl: " . . . and covered your nakedness. I pledged myself to you and entered into a covenant with you, says the Lord GOD, and you became mine" (v. 8a₂-b).[24] Like the image of Nebuchadnezzar stretching out his שפרירו (canopy/net, Jer 43:10, 12) over Egypt to apply his dominion over the land,[25] the covering of her bareness claims enduring ownership of the girl. However, within the specific familial metaphor, the covering also alludes to sexual commitment,[26] as is further indicated by the allusion to the blood rinsing off the girl (Ezek 16:9), possibly referring to her menstrual blood (and thus her "womanhood"),[27] and the so-called hymeneal blood, a result of her first sexual encounter.[28] Hence, the word דמים is in plural form. Thus, what started as Yhwh's parental care for the girl, has been replaced by a marital relationship enforced upon her.[29]

As has been outlined by several scholars, the allegory alludes to an incestuous relationship between Yhwh, an adoptive father, and his daughter, who is coerced into a wedlock.[30] In this, the allegory reflects a conviction that God is exempted of what is forbidden in human conduct.[31] The tendency of ascribing to the deity different morality also appears in biblical narratives describing or depicting God's unexplained wrath and retaliation (e.g., 1 Sam 6:19–20; 2 Sam 6:6–7), in contrast to the set of clear rules enforced in human courts (cf., Deut 24:16; Jer 31:29–30). In Ezekiel, God's destructive commandments (Ezek 20:25, 26) constitute the opposite conduct of what is expected of human individuals in society (cf. Exod 13:11–15; Lev 18:5).[32] The incestuous relationship is also another manifestation of this. Ruth Netzer finds a similar phenomenon in Greek mythology, where incestuous interaction is legitimately regarded in the pantheon, unlike in human families (Netzer 2020, p. 201).

As said previously, Ezekiel is not the first prophet to use metaphors of parenting and marriage in portraying the relationship between Israel and God.[33] However, the allegory in Chapter 16 is especially exceptional in its narrative format, its length, and its consistent focus on the parent's interest, assigning to the child an utterly passive role. Unlike imageries that attribute to the nation a role in recuperating the relationship (cf., Jer 3:19-22), in the Ezekiel allegory, the destiny of the relationship is solely dependent on Yhwh through all stages—the adoption, conversion, and restoration. The incest forced upon the child is a type of exploitation that protects the sovereignty of the parent.[34] The girl is needed for the parent's control and dominion, and for a recognition and validation of his grandiosity,[35] and so, in other words, she is the narcissists' source supply. The parent achieves sexual benefits, and ignores any harmful consequences to the child. As expected, the allegory offers no discomfort or remorse regarding the portrayal of an incestuous relationship. Instead, it reveals that the conversion of the relationship is self-serving for Yhwh, while the child gains nothing. The child, as we shall see, is suppressed under the confining cloak of a narcissistic father.

## 8. Suppressed by a Narcissist Father

There is no automatic connection between narcissism in leadership, and in a family; but in the case of Yhwh, as depicted in Ezekiel, the two phenomena coincide. Yhwh's conduct towards the adopted daughter intends to achieve her utter compliance, thus fulfilling his own needs of grandiosity.

Parents with NPD incline to perceive their children as a potential mirror of themselves, aspiring the children to fulfill their parents' visions and to grant them acknowledgement and admiration.[36] At first, Yhwh adorns the nation-girl with gifts (gold and silver jewelry and exquisite clothing) (Ezek. 16:11–13), and with provisions such as flour, honey, and oil (v. 13). These all aim to supplement the girl's beauty and make her "fit to be a queen" (v. 13); thus, increasing her fame and reputation " . . . among the nations on account of [her] beauty" (v. 14). However, the fame and beauty bestowed upon the girl ultimately served

to praise Yhwh's capacity and authority: " . . . for it [her beauty] was perfect because of my splendor that I had bestowed on you . . . " (v. 14). While at first, Yhwh's gestures to the girl were supposedly prompted by her agony and dependency, later on her agony was the very circumstances that enabled Yhwh to take advantage of her, dismissing the aspects of care and support provided. The initial rescue thus led to dominion, and physical care was used to manifest his own splendor.

Parents defined as narcissistic often demonstrate the tendency to force expectations upon the child and, as a result, experience anger and disappointment when they lose hold of them. For parents, a sign of the child's individualism and self-exploration is experienced as hurtful and infuriating. This propensity, in most cases, exposes the parent's own drama: an underlining insecurity regarding their power in the family or elsewhere.[37] Yhwh is seemingly self-assured when imposing demands upon the girl, but his anger and aggressiveness towards her indicate his concern of losing control and being deprived of love and appreciation.[38] The girl's failure to show gratitude and compliance evokes his acute rage: "And in all your abominations and your whorings you did not remember the days of your youth, when you were naked and bare, flailing about in your blood. After all your wickedness . . . " (vv. 22–23). A restoration of his confidence and self-control is later achieved by inflicting punitive measures upon the inferior girl.

In authoritative families, punishments are employed in order to consolidate power and authority. More than pedagogical tools, punishments aim to instill compliance and diminish self-esteem among targeted family members. Shifts between various types of retaliation (e.g., over-intervention, outbursts, physical abuse, emotional blackmail, anger, disregard, neglect) ensure submission to and dependence upon the powerful person.[39] In the allegory, sanctions are inflicted not directly by the parent, but by the girl's "lovers". God executes punishment by proxy, whom he takes "to do the dirty work". The girl is stripped, humiliated and bitten, in a violent gang rape (vv. 37–41), and thus is publicly returned to the state of "naked and bare" (v. 39), as she was before Yhwh took notice of her (vv. 4-5).[40] Not only does she lack the support and protection she deserves from her caregiver, but also loses any alternative she sought through betrayal, which became a source of violence and agony for her. This penalty is a radical manifestation of what Johanna Stiebert defines as the patriarchs' "honor–shame complex" concerning their daughter's behavior.[41] By having the girl violated and dishonored by others, Yhwh enforces upon the girl the same scenario he despised. By that, he amplifies the allegations against the girl, without admitting that it was him who had already crossed the line with her.

The execution of punishments calms Yhwh down and brings him to satisfaction. Upon achieving the girl's full submission, his confidence and tranquility are restored: "So, I will satisfy my fury on you, and my jealousy shall turn away from you; I will be calm, and will be angry no longer" (v. 42). As for the girl, she is left to bear shame and disgrace (vv. 54, 61), and becomes fully compliant henceforth, acknowledging Yhwh and nothing else (יהוה וידעת כי אני, "and you shall know that I am, the Lord", Ezek 16:62). David Blumenthal summarizes the narrative up to this stage as recounting God's taking the girl back in love after sexually abusing her.[42] It should be noted that this occurs only after watching her being violated and dishonored by others.

A child growing up in such circumstances does not have much of a choice. The allegory terminates with the proclamation that the covenant is reestablished, with no agency of the child or a new appreciation towards the caregiver, but rather as total submission, endured in complete silence: " . . . remember and be confounded, and never open your mouth again because of your shame, when I forgive you all that you have done, says the Lord GOD" (v. 63).[43] This submission, reaffirming the parent's grandiosity, is planned to last forever (ברית עולם, "an everlasting covenant", v. 60).[44]

Studies on narcissistic parenting demonstrate the various damaging consequences of such child rearing.[45] However, Yhwh (and the narrator) denies the girl's need for selfhood and regards her as the one to blame, rather than acknowledging her ordeal.[46] This

happens in the space where she mostly needs protection, a protection she cannot obtain elsewhere. Yhwh's narcissistic nature is the basis for the girl's tragic life, full of cruelty and maliciousness. Michael Coogan drew attention to this harmful relationship between Yhwh and Jerusalem in the allegory, in which the patriarch is "insanely jealous and abusive", who "subjects his wife to gang rape and gang murder, as with the Levite's concubine in Gibeah. Yet, unlike the Levite's concubine, no trace of sympathy is expressed for these wives of the deity".[47] Coogan's warning has been echoed by feminist scholars who saw in the allegory an endorsement of violence against women.[48] However, this textual evidence cannot merely be considered as misogyny. The girl/woman in the allegory signifies the larger congregation of abused men and women who receive no sympathy.

It is true that outlined here is a mere allegory, which cannot be taken literally. However, we should also admit that any depiction and representation of the deity in the Bible, and elsewhere, is an attempt to depict the abstract and unreachable. The sum of all allegorical attempts reflects the way people have perceived themselves, through a consideration of the form of the eternity and their relationship with it. With no self-reflection, or a stated criticism, this allegory constitutes an illustration of the supreme deity as inflicting malevolence upon his subjects. His behavior and inclination are comparable to that of humans, including those with a range of psychological complexes. Could we then contest the existence of evil in God's personality?

### 9. Evilness in God—Conclusions

We can now return to Block's argument concerning Yhwh's character in the book of Ezekiel, supposedly manifesting grace and compassion. This interpretation does not adhere to the image of God in theological statements spread throughout the book of Ezekiel, and it certainly does not fit the picture offered by the allegory in Chapter 16. Even if we put aside the harsh attitude towards Judah, both in the broader context and in the allegory (which may indeed be assessed differently by various scholars and generations)—we are left with God's own-declarations which deny any sign of "grace and compassion". The declarations disclose the purpose of God's interaction with the people, which is solely for the sake of God's validation and grandiosity. This determines his judgment and verdict upon the subjects—Israel/Judah/Jerusalem, which serve to amplify and disseminate his ultimate authority.

According to the declarations, Yhwh acts in consideration for his reputation and sanctification. This self-absorbed characteristic demonstrates a personality with narcissistic disorder.[49] The lack of grace and compassion of this character is especially conveyed through the framework of Yhwh as the people's patriarch, which admits an exploitation of the people for the sake of his strength and self-confidence. While this also reveals his insecurity, Yhwh knows how to compensate for it by controlling and terrorizing his kin. The narcissistic character of Yhwh is expressed to the fullest when dealing with those who need his protection most. His character as a supreme authority cannot be defined without acknowledging these elements, which are widely accepted as evil and cruel.

**Funding:** This research received no external funding.

**Institutional Review Board Statement:** Not applicable.

**Informed Consent Statement:** Not applicable.

**Data Availability Statement:** Not applicable.

**Acknowledgments:** Thanks to Ohad Magori, Shlomith Cohen, Dan'el Kahn, Itzhak Feder, and the three anonymous referees, for helpful advice on earlier drafts of the article. The article is in the memory of Lucy Davey (1940–2022, Sydney), who recognized evil, and chose good.

**Conflicts of Interest:** The author declares no conflict of interest.

## Notes

1    Leibniz, one of the rationalistic pioneers on the subject, suggested that God, as an omnipotent, omniscient, and omnibenevolent, created the "Best of all possible worlds". The evils that people do, even out of free will, or experience and witness, are such only due to their humanly limited understanding, as they are part of the grand formula leading to such success. See (Leibniz [1710] 2007).

2    (Sterba 2019, pp. 134–5, 190–92). See also (Wilmot 2021).

3    See (Crenshaw 1983, pp. 1–5; Swinburne 1983; Davies 2006, p. 154).

4    Another theistic suggestion is that evil is a result of the free will given to humans, and which humans can practice and control. According to Eleonore Stump, the possession of free will is such a good value (both in Christianity and in general) that it outweighs all evil in the world (Stump 1985, p. 416).

5    The link created by theists and philosophers between omnipotence and goodness is that of a deductive nature. Origen of Alexandria explains that by definition, God's qualities are not absolute, whereas he cannot act out 'any' action as his actions are limited to absolute goodness, justice and wisdom (Origen 1885). John Mackie discusses the paradox existing between the three suppositions: God is omnipotent, God is wholly, evil exists, suggesting that never can the three coexist (Mackie 1955). McCloskey asserts that the existence of evil implies either that there is no God or that the "god" who exists is imperfect either in power or in goodness (McCloskey 1962).

6    Hobbes asserted that primal human nature is violent and self-serving, seeking dominance over others. He stated that *good* and *evil* are defined subjectively through personal desire or aversion (Hobbes 1939, pp. 149–50). Rousseau in contrast, believed that primal human nature was based on self-love, which did not lead to violence: "There is, then, deep in our souls an inborn principle of justice and virtue by which, in spite of our maxims, we judge our actions and those of others as good or bad; and it is to this principle that I give the name of conscience" (Rousseau [1762] 1957, p. 40).

7    Indeed, the punitive situation of the people of Israel has been supposedly known to the other nations, as they testify: "these are the people of the LORD, and yet they had to go out of his land" (36:20. Cf., 39:23, in the case of "the house of Israel").

8    And see: ונחלת בך לעיני גוים (22:16, MT: second person: "and you [sg. f.] shall be profaned"). Several ancient versions read the verb as first person, in which case the Lord refers to how his people's sin brings disgrace upon him: "And I shall be profaned through you in the sight of the nations; and you shall know that I am the LORD" (Ezek. 22:16). For a defense of the MT, see (Block 1997; Greenberg 1983, pp. 457–58).

9    Barker surveys occurrences of all the definitions stated by Dawkins, and adds further traits, under the title "Dawkins Was Too Kind", such as Pyromaniacal, Angry, Merciless, Curse Hurling, Vaccicidal, Aborticidal, Cannibalistic, Slavemonger; and he does not spare the image of Jesus from this.

10   See for example (Jung [1954] 2002) (e.g., "Yahweh had one good son and the one who was a failure. Both Cain and Abel and Jacob and Esau correspond to this prototype, as does the motif of the hostile brothers in all ages and all parts of the world. Innumerable modern variants cause dissension in modern families and keep the psychotherapist busy", p. 38). (Morrow 2004), e.g., "There is evidence that the Babylonian exiles exhibited psychological symptoms known among groups of persons displaced by violent processes . . . The exiles were burdened with low esteem for the faith community called Israel to which they and the previous generation, which had actually suffered through the violence of the Babylonian conquest and deportation . . . ", p. 85); (Abramovitch 2014) (e.g., "In describing the birth of the first brothers, these opening two verses of Genesis 4 reveal much about the psychodynamics of birth order . . . Firstborns must live up to intense parental projections. Eve does not say, 'I have gotten a child', but 'I have gotten a man'. Eve does not see the child, only the man he is to become", p. 29); (Markl 2020) (e.g., "While reflection on the psychological background of ancient texts is necessarily hypothetical and speculative, trauma theory may help explain the rhetoric of blaming and shaming employed in the Song of Moses at the culmination of the Pentateuch. The Song may be understood as an intellectually worked through externalisation of self-blame and shame, an elaborate expression of cultural trauma", p. 686).

11   (Capps 2009, p. 195). Capps mainly based his demonstration on texts from Genesis, Exodus and Job.

12   (Capps 2009, pp. 200–204). These are the nine characteristics: 1. Has a grandiose sense of self-importance. 2. Is preoccupied with fantasies of unlimited success, power, brilliance, beauty, or ideal love. 3. Believes that he is "special" and unique and can only be understood by, or should associate with, other special or high-status people (or institutions). 4. Requires excessive admiration. 5. Has a sense of entitlement 6. Is interpersonally exploitative. 7. Lacks empathy: is unwilling to recognize or identify with the feelings and needs of others. 8. Is often envious of others or believes that others are envious of him or her. 9. Shows arrogant, haughty behaviors or attitudes.

13   (Hallford and Linebach 2021, p. 27); See also (Peck 1986) stating that "In addition to the fact that the evil need victims to sacrifice to their narcissism, their narcissism permits them to ignore the humanity of their victims as well" (p. 136).

14   In 1964 Erich Fromm coined the term "Malignant Narcissism" to define the syndrome of the extreme mix of narcissism, antisocial behavior, aggression and sadism, stating that it represents "the quintessence of evil" (Fromm 1964, p. 37). Since then it has been widely accepted that malignant narcissism is a form of highly abusive and manipulative NPD. See (Shafti 2020).

15   Not always, however, the "daughter" metaphor in the prophetic discourse communicates a relationship of loyalty and friendship. A common use of the metaphor is for highlighting the nation's negative traits: disloyalty (e.g., Isa 1:21; Jer 2:20; 13:22–7) and defeat (e.g., Isa 47). This tendency makes the daughter imagery close to the metaphoric usage of the treacherous wife and harlot, thus evoking the husband's legitimacy to inflict punitive measures.

16   (Block 1997, p. 13). Block cites (Halperin 1993, pp. 170–71).

17   Cf., Moshe Greenberg's suggestion that the act of stripping an adulterous female was an ancient judicial practice in Israel, as indicated in Hos 2:12; Nah 3:5; Jer 13:22, 26 (Greenberg 1983, p. 286). Daniel Smith-Christopher disagrees and argues that these illustrations derive from legal *divorce ceremonies* rather than public trials or punishments for adultery (Smith-Christopher 2004, pp. 144–46). Galambush recognizes here a multilayered exposure similar to that achieved in cinema through the "male hero's gaze" which controls the spectator's view of the woman (Galambush 1992, p. 94).

18   Elsewhere, Ezekiel seems to be familiar with the tradition of the Patriarchs (cf., 33:24), but opposes it in this context. This is possibly part of the attempt to emphasize the role of God in redeeming Israel from her low and condemned status. Nonetheless, this rhetoric may also derive from ethnic data, attested in other places, such as the tradition of Jerusalem's Jebusite (Canaanite) origins (2 Sam 5:6–8), and the tradition that the Canaanite population had continued to reside in the land, alongside the Israelites. For a discussion of the purpose of this fictive genealogy and of the way it was understood by medieval Jewish exegetes, see (Rom-Shiloni 2011, pp. 99–103).

19   See Malul's compelling proposal that the portrayal of the failure to wash and feed the infant signifies parental denial of legal recognition: (Malul 1990, p. 109). For a broader discussion of the practices that were deprived of the newborn girl (cutting her umbilical cord, washing her, rubbing with salt, or swaddling her), see (Philip 2006, p. 95).

20   (Maier 2008, p. 115). The assumption that God did not actually do anything for the infant was also pointed out by (Day 2000, p. 207). See also Halperin, who says: "So little 'nurturant' is Ezekiel's God that it does not occur to him so much as to bathe the girl until he is ready to take her to bed (verse 9)" (Halperin 1993, p. 173).

21   Malul, "Adoption of Foundlings in the Bible and Mesopotamian Documents" (Malul 1990, p. 111).

22   Unlike the MT (*Masoretic Text*, Hebrew manuscript) that mentions Yhwh's action in the expansion of the girl (רבבה כצמח השדה נתתיך), the NRSV translation indicates no involvement of Yhwh in the fulfilment of the girl's growth: "and grow up like a plant of the field" (v. 7a). This contrasts not only the MT version (with the verb נתתיך, I gave/made you), but also the LXX (Greek version), which states: "πληθύνου καθὼς ἡ ἀνατολὴ τοῦ ἀγροῦ δέδωκά σε … " (δέδωκά σε, "I gave you").

23   See Runions' argument about the allegory: "to read the relationship between the woman and the deity as a sexual relationship is either to ignore the obvious parental imagery of vv. 1–13 or to tacitly condone incest" (Runions 2001, p. 160).

24   See Pardes: "the prophetic preoccupation with female nakedness (Ephraim, the male personification of the nation is never uncovered) seems to exhibit an all too common patriarchal need to control women's bodies and women's sexuality … to make clear distinctions between women whose bodies are owned by given men (father, brother, or husband) and those that may be regarded as public property. A woman who does not maintain her nakedness under cover exposes herself to the danger of being undressed in public" (Pardes 1992, pp. 134–35).

25   On the role of this motif in Mesopotamian literature and its influence on biblical imagery see (Goldstein 2020, pp. 63–76).

26   See Howard Eilberg-Schwartz, about the act of Yhwh's spreading his cloak "as close as we get to a graphic image of God having sexual intercourse" (Eilberg-Schwartz 1994, pp. 111, 113). This may be supported by the allusion to Ruth's request of Boaz to spread his cloak (Ruth 3:8–9) as a supposed euphemism for sex, within the paradigm of a "legitimate intercourse" occurring "under covers" (cf. Hos 2:11). See (Kruger 1984, p. 86; Pardes, *Countertraditions in the Bible*, p. 134).

27   Cf., the NRSV translation for עדי עדים (v. 7, "ornaments of ornaments") as "full womanhood", implying the idea of arriving at "the time of menstruation", the age of sexual engagement.

28   To these two layers of blood Koller adds the "birth blood in which the girl has been wallowing for more than a decade" (and was not taken care of, namely was not adopted, according to his thesis) (Koller, "Pornography or Theology?", p. 411. See also Greenberg 1983, p. 278). For a discussion regarding the various types of feminine bloods see (Philip 2006, pp. 66–67).

29   There is no romantism or a mutual choice here (unlike the one can be detected in the tale of Ruth and Boaz, for example, or in the images of the Song of Songs, where the woman even initiates the intimacy).

30   See scholars calling attention to the disturbing image of Yhwh in Ezek 16:8, playing a foster father having sexual relations with his foster daughter: (Seifert 1997, pp. 262–68; Baumann 2003, p. 161; Galambush, *Jerusalem in the Book of Ezekiel*, p. 94).

31   It is worth noting, however, that the father-daughter connection is strikingly absent in the list of laws that prohibit incestuous relationships (Leviticus 18, 20). The absence of an explicit prohibition of father-daughter incest reveals an ambiguity about this practice, and the possibility that at times, this type of relationship was tolerated. See (Cardascia 1980; Frymer-Kensky 1992, p. 1145; Carmichael 1995, pp. 127–28; Ziskind 1996). Nonetheless, despite the lack of explicit prohibition, it is reasonable that fathers had abode by society's expectations and did not sexually abuse their daughters. In the words of Jonathan Ziskind, the fathers were "mindful of the social and financial advantages of offering to a prospective son-in-law a daughter who was a virgin" (Ziskind 1996, p. 130).

32  Julian Pitt-Rivers has demonstrated this tendency in the Hebrew Bible, in which "pure myths" reflect values that are contradictory to what is culturally accepted. See (Pitt-Rivers 1977, pp. 151–55). See also (Kugler 2017, pp. 54–56).

33  These themes are found in pre-exilic prophecy (e.g., Is 1:21; Jer 2:2, 3:2–10, 20; Hos 2:4–25), some of which may be the textual basis and inspiration for the lengthy allegory of Ezekiel (esp. Isa 1:21 and Hosea 2). See (Cooke 1937, p. 159; Wolff 1975, pp. 12–17, 30–37, 70–93; Setel 1985; Bird 1989, pp. 88–89).

34  See (Kinnear 2007, p. 8), about the reasons that drive perpetrators of incest. It may be an intense sexual desire for children (pedophilia), which is carried out upon those most available to them, their children. Or perpetrators are driven by the attempt to fulfil their sexual needs and fantasies without "harming the family" by conducting relationships outside of the family.

35  See the findings of Herman about the fragility and lack of security of perpetrators in incestuous relationships: "Other observers . . . have described the same fathers as 'ineffectual and dependent,' 'inadequate,' or 'weak, insecure and vulnerable.' Far from appearing as tyrants, these fathers emerge as rather pitiful men, sometimes even as victims of a 'domineering or managing wife'" (Herman 2000, p. 74).

36  See: (Hendrick 2016, pp. 4–5, 22–23; Brown 2020, pp. 1–22); See also a website article: (Banschick 2013).

37  See (Myers and Zeigler-Hill 2012).

38  See Barker on God's behavior, stating "Look at me! I am the great and terrible Lord!": "It seems to me that a truly great person would not have to brag about it. Truly great people don't need to draw attention to themselves. A truly great person is concerned about the effects of their actions in the real world, not about how they are perceived by underlings. Truly great people are psychologically secure, not dependent on the opinions of others. God is not great. He is merely megalomaniacal" (Barker 2016, *God: The most unpleasant character in all fiction*, p. 221).

39  See (Gardner 2004; Rappoport 2005; Brown 2020, pp. 59–60).

40  Ralph Klein refers to the role of "nakedness" in vv. 37–39 as a "negative inclusio" with the birth narrative of vv. 4–6 (see (Klein 1988, p. 88).

41  (Stiebert 2013, p. 189).

42  (Blumenthal 1993, p. 241). Furthermore, he emphasizes that "What is true of abusive behavior by humans is true of abusive behavior by God. When God acts abusively, we are the victims, we are innocent . . . the reasons for God's actions are irrelevant, God's motives are not the issue. Abuse is unjustified, in God as well as human beings". (here p. 248). It is his opinion that an abused child must come to terms with the abusing parent, like Israel with their abusive God.

43  Cf. the situation where a girl's silence during intercourse outside of marriage (including when it is forced on her) is criticized and sentenced with a death penalty: Deut. 22:23–24. In Ezek 16:63, Greenberg's use of "absolve" instead of the NRSV "forgive" renders the Hebrew כפר more accurately (Greenberg 1983, pp. 273, 291). Jon Levenson identifies here "restoration [that] replaces retribution" (Levenson 2015, p. 120).

44  The other nations as well, according to Ezekiel's theology, fulfil God's goal of being recognized and known, by experiencing his divine wrath and envy in a similar way to Israel. For example, Ezekiel says in regard to Mount Seir (Edom): "therefore, as I live, says the Lord God, I will deal with you according to the anger and envy that you showed because of your hatred against them; and I will make myself known among you, when I judge you" (Ezek. 35:11), and Gog: "With pestilence and bloodshed I will enter into judgment with him; and I will pour down torrential rains and hailstones, fire and sulfur, upon him and his troops and the many peoples that are with him. So I will display my greatness and my holiness and make myself known in the eyes of many nations. Then they shall know that I am the LORD" (38:22–23).

45  See (Manzano et al. 2005, pp. 117, 141–49).

46  Cf., the reading of Levenson of this passage, which seems as identifying with the perspective of the narcissistic father: "Forgetful of her humble origins and of her husband's generosity as well, God's metaphorical wife has lost all sense of her dependence on him" (Levenson 2015, p. 120).

47  (Coogan 2011, pp. 186–87).

48  See Mary Daly on God's maleness as legitimates oppression and abuse of women: (Daly 1986, pp. 98–101). Fokkelien van Dijk-Hemmes argued that both Ezekiel 16 and 23 encourage abuse of girl-children: (van Dijk-Hemmes 1995).

49  For the term "self-absorption" as a narcissistic characteristic, see (Brown 2020, pp. 14–17).

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
