# Peer review of "“And You Became Mine” (Ezek 16:8): Good and Evil in a Narcissistic God"

_religions, doi:10.3390/rel13100967_

Round 1

Reviewer 1 Report

It seems to me that there is a logical problem with your thesis (81-83).  I don't think that there is a supposition that omnipotence presupposes only good.  In fact, there are plenty of examples to the contrary.  Leibniz, for example, distinguishes omnipotence from benevolence.

Your second thesis statement (89-92) is persuasive, but it doesn't seem to me to be very original.

I don't find it persuasive, personally, that a deity can be diagnosed with a personality disorder; however, it is a thesis, and theses are meant to be disagreed with.  The evidence provided is sufficient.

The interpretation of Ezekiel was, for me, the most interesting part of the article.

Overall, although the English is impressive, at certain points in the article it is clear that the author is not a native speaker.  This struck me as soon as I read the abstract.  A well-educated native speaker needs to go over and revise the entire manuscript.

Author Response

thanks very much. 

Reviewer 2 Report

This is a well-written paper, identifying elements of narcissism in the image of God as depicted in religious texts. Should the Editor accept it, the paper will be well placed in Religions journal.

My recommendation is Accept with major/moderate Revisions.

My concerns with this paper revolve around style and readability. First, the reader has to do a lot of deciphering to get to the argument, especially in the first four sections of the paper. More precisely, the first two sections seem to be preoccupied with the main questions of the present study,  but these questions are not clearly illustrated. They are interrupted by arguments and counter-arguments and the reader is not able to get a precise view of what steps the authors will take, how they will end up to the main conclusion about God and narcissism. Section three starts immediately without providing solid explanations about the arguments that will be developed there. I suggest the authors premise the Introduction with a clear rationale for the paper, outline their argument, and situate it within a broader literature. At the end of each section it would be important to state clearly what will be discussed in the next section and for what reason. For example, 'in this section the A has been made clear. We will move ahead discussing the B, which is important in order to end up to a conclusion concerning D'.  Consider, for example, the end of the fifth section: 'As we shall see, Yhwh of the allegory is driven by his own needs, and not by the needs of his adopted child/spouse. This correlates with the broader view of God in Ezekiel, acting for the sake of himself, with no consideration for humans, including his very close kin, Israel. Traits such as these are rudimentary in NPD.' This is a good example of signposting. Here the reader understands what the next section is about and what is the practical contribution of the discussion that will take place there. 

Second, many arguments expressed in this paper make quick, sweeping claims and shortcuts pile up to lead to categorial assumptions. Again, this is more evident in the first five sections. For example, '[a] primary criterion for examining people's traits of "goodness" and "evil" is through an evaluation of their relationships with others, and their attitude toward them, revealing intentions of good will or inclination to cause harm.' It would be better to avoid using such an absolute tone by re-phrasing the sentence as follows: 'one could acquire the primary criterion for examining people's traits of "goodness" and "evil" through an evaluation of their relationships with others, and their attitude toward them, revealing intentions of good will or inclination to cause harm.' In addition, '[a] compelling illustration of God's association with the chosen Israel is narrated in the book of Ezekiel'. It seems that you are taking this is as a truth, which (in fact) is a debatable argument. One could, therefore, ask: is this the only image of God illustrated in the book of Ezekiel? It would be, therefore, important to stress that such an illustration of God one could acquire in the book of Ezekiel. Thereupon, justify the argument. In this way, the reader sees a balanced, nuanced and moderated view, giving no impression that agreement with the interpretations presented by the authors is inevitable. Likewise, '[a] tumultuous allegorical narrative in Ezekiel displays a dark picture of a treacherous and violent relationship between an authoritarian entity, Yhwh, and the helpless dependent female child, Jerusalem. Yhwh, a patriarchal character, plays dual roles for the female entity: first a caregiver, then a husband.' Without disputing the validity of this statement, one could ask: why does it play dual role for the female entity? Why 'dark' and 'authoritarian'? Maybe you could suggest that 'our interpretation ... will demonstrate that in Ezekiel the relationship between Yhwh and the child is authoritarian', providing explanations that gratify this statement. 

Third, the paper needs deeper analysis on sources. Some of the conclusions constructed upon interpretation of religious texts are too quick and broad. It would be important to develop your arguments further, write a few sentences (if necessary). Finally, the authors need to be aware of the fact that some paragraphs and sentences seem to follow a different writing style. It is evident that they are written by different authors. 

In general terms, I do think that the paper should be published. However, the argument needs revising in order to look stronger (better justified), better nuanced and moderated. In addition, the paper needs a clear structure, with all the steps that will be followed explained from the beginning. At the same time, signposting sentences are needed between each section in order to alert the reader about discussions that follow. This will remarkably improve the flow. 

Author Response

Thanks very much. 

Reviewer 3 Report

Besides some typos (detailed below), I see no problem with this paper. The author is very informed and has the necessary expertise to make the argument--and they make it well. There needs to be more papers like this, that are brutally honest about what the god of the HB/OT is like (rather than those, like Block, who just make up ad-hoc rationalizations for ignoring the obvious meaning and instead reading the text in a way that comports it with modern assumptions and sensibilities).

I’d love to see a paper, by this author, that talks about the same kind of problem extensively utilizing Ezekiel 25:17 (and the surrounding passages); because that passage was made famous by Pulp Fiction (although Tarantino greatly modified the verse), I feel like that article might actually catch public attention, and thus bring the public eye to a very needed honest articulation of how the HB/OT depicts its god.

Line 86 – typo: delete the word “is.”

Line 123 – the grammar here is confused. Reword.

Line 124 – extra space

Line 131 – add an oxford comma

Line 163 – add an oxford comma

194 – the author might think of add the story of Job as an example of such behavior.

310 – I think it is supposed to read “is plural” not “in plural.”

426 – I think there is an extra it. it should be “… particularly does not fit…”

Author Response

Thanks very much 

Round 2

Reviewer 1 Report

I see that the authors have thoroughly revised the paper.  They have made all of the substantive changes I suggested.  However, it does need one last proofreading for typos and stylistic details.